# A Spanish Family with Gordon Syndrome Due to a Variant in the Acidic Motif of *WNK1*

**DOI:** 10.3390/genes14101878

**Published:** 2023-09-27

**Authors:** Ramón Peces, Carlos Peces, Laura Espinosa, Rocío Mena, Carolina Blanco, Jair Tenorio-Castaño, Pablo Lapunzina, Julián Nevado

**Affiliations:** 1Department of Nephrology, Hospital Universitario La Paz, IdiPAZ, Universidad Autónoma, 28046 Madrid, Spain; 2Area de Tecnología de la Información, SESCAM, 45003 Toledo, Spain; cpeces@externas.sescam.jccm.es; 3Department of Pediatric Nephrology, Hospital Universitario La Paz, IdiPAZ, Universidad Autónoma, 28046 Madrid, Spain; laura.espinosa@salud.madrid.org; 4INGEMM, Institute of Medical and Molecular Genetics, La Paz University Hospital, IdiPAZ, 28046 Madrid, Spain; mariarocio.mena@salud.madrid.org (R.M.); carolina.blanco.agudo@gmail.com (C.B.); jaira.tenorio@salud.madrid.org (J.T.-C.); p.lapunzina@gmail.com (P.L.); 5ITHACA, European Research Network, La Paz University Hospital, 28046 Madrid, Spain; 6Network for Biomedical Research on Rare Diseases (CIBERER), Carlos III Health Institute (ISCIII), 28046 Madrid, Spain

**Keywords:** familial hyperkalemic hypertension, Gordon syndrome, pseudohypoaldosteronism type II, metabolic acidosis, *WNK1* missense variant

## Abstract

(1) Background: Gordon syndrome (GS) or familial hyperkalemic hypertension is caused by pathogenic variants in the genes *WNK1*, *WNK4*, *KLHL3,* and *CUL3*. Patients presented with hypertension, hyperkalemia despite average glomerular filtration rate, hyperchloremic metabolic acidosis, and suppressed plasma renin (PR) activity with normal plasma aldosterone (PA) and sometimes failure to thrive. GS is a heterogeneous genetic syndrome, ranging from severe cases in childhood to mild and sometimes asymptomatic cases in mid-adulthood. (2) Methods: We report here a sizeable Spanish family of six patients (four adults and two children) with GS. (3) Results: They carry a novel heterozygous missense variant in exon 7 of *WNK1* (p.Glu630Gly). The clinical presentation in the four adults consisted of hypertension (superimposed pre-eclampsia in two cases), hyperkalemia, short stature with low body weight, and isolated hyperkalemia in both children. All patients also presented mild hyperchloremic metabolic acidosis and low PR activity with normal PA levels. Abnormal laboratory findings and hypertension were normalized by dietary salt restriction and low doses of thiazide or indapamide retard. (4) Conclusions: This is the first Spanish family with GS with a novel heterozygous missense variant in *WNK1* (p.Glu630Gly) in the region containing the highly conserved acidic motif, which is showing a relatively mild phenotype, and adults diagnosed in mild adulthood. These data support the importance of missense variants in the *WNK1* acidic domain in electrolyte balance/metabolism. In addition, findings in this family also suggest that indapamide retard or thiazide may be an adequate long-standing treatment for GS.

## 1. Introduction

The Gordon syndrome (GS), also known as familial hyperkalemic hypertension (FHH), chloride shunt syndrome, or pseudo-hypoaldosteronism type II (PHAII) (MIM#145260), is an infrequent disorder associated with net positive Na^+^ balance and renal K^+^ retention resulting in hypertension, hyperkalemia, and hyperchloremic metabolic acidosis, low plasma renin (PR) activity, and usually normal plasma aldosterone (PA) levels. It is commonly associated with average glomerular filtration rate (GFR) and high responsiveness to thiazides and dietary sodium restriction [1,2,3,4]. GS can be caused by dominant pathogenic or likely pathogenic variants in the genes *WNK1* (OMIM # 605232) on 12p13.33, *WNK4* (OMIM # 601844) on 17q21.2, *CUL3* (OMIM # 603136) on 2q36.2, and either dominant or recessive pathogenic or likely pathogenic variants in *KLHL3* (OMIM # 605775) on 5q31.2 [5,6]. In addition, there is not much information on the long-term prognosis of patients with GS. There is a genotype–phenotype correlation for the genes associated with GS. Thus, patients with heterozygous *CUL3* pathogenic variants are more likely to develop severe hyperkalemia, metabolic acidosis, hypertension, and growth impairment at an early stage than individuals carrying recessive *KLHL3*, dominant *KLHL3*, *WNK4*, or *WNK1* pathogenic variants [6]. A milder phenotype for the *WNK1*-based individuals has been suggested, which could be partly explained by a different pathogenic mechanism [5,6,7]. Thus, *WNK1* exon 7 missense variants leads to hyperkalemic/hyperchloremic metabolic acidosis with normal blood pressure in most of the affected patients [7].

Here, we present detailed clinical and molecular findings of a large Spanish family with GS carrying a novel, heterozygous, likely-pathogenic missense variant in *WNK1,* located within exon 7. In addition, we also present the first series of GS in relation to the long-term follow-up of those successfully treated with a low dose of indapamide retard.

## 2. Materials and Methods

### 2.1. Biochemical and Radiological Studies

The pedigree of the family with GS is described in Figure 1. Only patients III-1, III-3, III-5, III-8, IV-3, and IV-6 have been examined (Figure 1).

We evaluated height, weight, blood pressure, and blood and urinary biochemical parameters in members of the family examined. Urine samples were collected for urinalysis: first-voided morning urine and 24 h collection. Serial measurements of serum and urine levels of Na^+^, K^+^, Cl^−^, HCO_3_^−^, creatinine, and other biochemical values such as estimated glomerular filtration rate (eGFR), FE_K_^+^, FE_Na_^+,^ and FE_Cl_^−^ were measured or calculated using standard procedures. The eGFR was determined by the chronic kidney disease epidemiology collaboration (CKD-EPI) equation using serum creatinine. PR activity (normal range: 1.5–5.7 ng/mL/h) and PA levels (normal range: 40–310 pg/mL) were determined using standard procedures. The clinical and laboratory diagnosis of GS was established by the presence of hypertension and/or hyperkalemia, hyperchloremia, metabolic acidosis, FE_K_^+^ ≤ 6%, FE_Na_^+^ ≤ 1%, and FE_Cl_^−^ ≤ 1.2%. Patients were evaluated for nephrocalcinosis, nephrolithiasis, or other renal abnormalities by abdominal ultrasonography (US). 

### 2.2. Genetic Studies

#### Massive Parallel Sequencing (NGS Analysis)

We used a custom-targeted NGS gene panel (NEFROseq v2.1) for the study of the full spectrum of genetic nephropathies, which contains 380 genes including the genes responsible for GS (*WNK1*, *WNK4*, *KLHL3*, and *CUL3*). Sequences were captured using SeqCap EZ technology (Roche Nimblegen. Madison, WI, USA) and subsequently sequenced on a NextSeq500 (Illumina. San Diego, CA, USA). Bioinformatics analysis was performed in our in-house Bioinformatic section applying publicly available software tools (trimmomatic-0.32; Bowtie2 v2.1.0; Picard-tools v1.27; Samtools v0.1.19-44428cd; Bedtools v2.26.0; Genome Analysis TK v3.3-0 y SnpE 4.1l; ClinVar date 20140703 dbscSNV1.1 dbNSFP version 3.0 dbSNP v138), and simultaneously using the variant Caller V2.1 tool (Illumina). In silico pathogenicity prediction was analyzed with Alamut^TM^ visual (SOPHIA GENETICS, SAS, Bidart, France) and other tools such as SIFT ensembl 66 (SIFT score range from 0 to 1. The smaller the score the more likely the SNP has a damaging effect; damaging < 0.06); Polyphen-2 v2.2.2 (the score ranges from 0 to 1; benign < 0.03); Mutation Assessor, release 2 (The score ranges from −5.17 to 6.49 in dbNSFP; damaging > 1.8); FATHMM-MKL, v2.3 (Scores range from 0 to 1. SNVs with scores > 0.5 are predicted to be deleterious, and those <0.5 are predicted to be neutral or benign. Scores close to 0 or 1 are with the highest-confidence); Gerp2 version 2010 (It ranges from −12.3 to 6.17, with 6.17 being the most conserved); PhyloP100way (scores are based on multiple alignments of 99 vertebrate genome sequences to the human genome. The greater the score, the more conserved the site); CADD, v1.3. (Scores above 20 are predicted to be among the 1.0% most deleterious possible substitutions in the human genome); DANN, v2014 (Scores range from 0 to 1. A larger number indicates a higher probability of being damaging). The allele frequency threshold used for variant filtering was 0.01 to avoid polymorphism higher than 1%. Population frequencies of the detected variants were assessed using the gnomAD exomes, gnomAD genomes, Bravo, Beacon, and 1000 genome project; Spanish Exon Variant Project; NHLBI exome sequencing project: ESP6500_EA_AF. Variants were classified according to the ACMG/AMP guidelines [8]. Candidate variants and segregation analysis were subsequently confirmed using Sanger sequencing. GRCh37 (hg19) was used as a human genome of reference.

## 3. Results

### 3.1. Patient’s Follow-Up

The family reported herein included six affected individuals (all with hyperkalemia and mild hyperchloremic metabolic acidosis despite a normal eGFR) (Figure 1 and Table 1). All adult women (III-1, III-3, III-5, III-8) presented with hypertension (superimposed preeclampsia in two of them), short stature, and low body weight, and the two children (IV-3 and IV-6) presented with normal height and weight for their age in the absence of hypertension. Laboratory findings also showed, in all cases, low FE_K_^+^, FE_Na_^+^, and FE_Cl_^−^ values, and low PR activity with average or low PA values (Table 1). The kidney US showed neither anatomical lesions nor calcification.

The index patient (III-1), at the age of 23 years, presented with a pregnancy that was complicated due to hypertension and superimposed preeclampsia. She also developed hemolytic uremic syndrome (HUS) with acute renal failure, requiring hemodialysis treatment during the third trimester of the pregnancy. After renal function recovery, hyperkalemic tubular acidosis was diagnosed, and oral treatment with sodium bicarbonate (500 mg three times a day) was started. The patient required a cesarean section and had a healthy boy with a low weight at birth. Afterward, dietary salt restriction and a low dose of thiazide were started. Since 2012, the treatment has been changed to indapamide retard 1.5 mg/day. 

Patient III-1 was the first child of an unrelated couple (Figure 1). Her father died at the age of 58 due to complications of hypertension and renal disorder. Her mother is alive (79 years old) without a history of GS or any other renal disease. Three of her four sisters were also affected based on medical records of hypertension and hyperkalemic tubular acidosis. Her first sister (III-3) had an obstetric history of two normal pregnancies with deliveries at the ages of 26 and 29 years old, respectively. However, at the age of 30 years, she presented with hypertension, hyperkalemia, and mild hyperchloremic metabolic acidosis. III-3 individual received dietary salt restriction, a low dose of thiazide, and sodium bicarbonate (500 mg three times a day) since 2012, and her treatment was also changed to indapamide retard 1.5 mg/day. Her second son (IV-3) was diagnosed with an evident hyperkalemic and mild hyperchloremic metabolic acidosis in the absence of hypertension with normal growth at the age of 10 years. The patient was lost to follow-up until the age of 27, and, since then, he has been treated with dietary salt restriction and indapamide retard 1.5 mg/day. The second sister (III-5), at the age of 31 years, presented with hypertension, hyperkalemia, and mild hyperchloremic metabolic acidosis. She received a low dose of thiazide and sodium bicarbonate (500 mg three times a day), and, since 2012, the treatment has also been replaced with indapamide retard 1.5 mg/day. Her younger sister (III-8) had an obstetric history of two abortions at the ages of 40 and 42 years, respectively. At the age of 43 years, she had a third pregnancy that was complicated due to superimposed preeclampsia, and she presented with hyperkalemia and mild hyperchloremic metabolic acidosis. Strikingly, after receiving dietary salt restriction, a low dose of thiazide, and sodium bicarbonate (500 mg three times a day), the patient had a successful full-term pregnancy (at 38 weeks of gestation) and delivery of a healthy girl with 2.6 kg of weight (percentiles: p10–p25). Since 2013, the treatment has been replaced with indapamide retard 1.5 mg/day. Her daughter (IV-8) presented with hyperkalemia in the absence of hypertension in the first three months of life and was then treated with dietary salt restriction and a low dose of thiazide. Despite the early manifestations, the administration of thiazide alone was sufficient to ensure relatively normal development and child growth (at the age of 5 years: body weight 19 kg, height 104 cm; 92nd percentile). The rest of the patients were able to tolerate indapamide retard treatment, which successfully normalized hypertension and reduced their concentrations of potassium to normality. Currently, all patients have been followed for >10 years and remain in good health on treatment. 

### 3.2. Genetic Findings

Genetic testing using a custom NGS panel (NEFROseq v2.1) revealed a novel heterozygous variant in the *WKN1* gene, NM_001184985.1:c.1889A>G; p.Glu630Gly (E630G), in the proband (Figure 2). Sanger sequencing confirmed the presence of the variant. 

This missense *WNK1* variant (Glu630Gly) is located within exon 7 of the gene, which encodes a highly conserved acidic motif in the protein, which has been previously shown to mediate the interaction with the substrate adaptor KLHL3 [7]. It was not reported previously in the literature nor in different specialized databases (LOVD v3.0; ClinVar, HGMD; accessed 20 June 2023). This variant is also not present in the analyzed pseudo-control population databases (absent in the gnomAD exome and genome database; https://gnomad.broadinstitute.org/; accessed 20 May 2023). Finally, this variant was classified using ACMG/AMP [8] as VUS with minor pathogenic evidence; PM2, PM5, PP3 (4 points pathogenic/0 points benign), and was predicted to be likely pathogenic by several in silico bioinformatic tools, see Table 2, and by family segregation. Genetic analysis of individual III-6 using Sanger sequencing did not show the variant. No other clinical significance variants were found in the other genes associated with GS.

Interestingly, the variant segregates in all affected individuals of the kindred (analyzed using Sanger sequencing) seemed the most likely cause of GS in this family. Altogether, we classify this variant as likely pathogenic.

Figure 3 shows a schematic representation of the WKN1 protein with conserved domains and all heterozygous missense variants reported for GS so far. Most of the missense variants mapped in the acidic motif are at exon 7 of *WNK1.*

## 4. Discussion

Here, we report a novel variant in the highly conserved acidic motif of the *WINK1* gene in a family of six members, and its response to thiazide/indapamide. In this variant, the amino acid change is a glutamic acid (E) to glycine (G) at codon 630, establishing the substitution of a negative glutamic acid by a neutral glycine in a highly conserved acidic motif of *WNK1*, whereas other missense variants have already been reported in patients affected by autosomal dominant PHAII [7,9,10,11]. In fact, other changes in the same amino acid (position 630) have also been associated with GS (Glu630Lys) [9,10]. To the best of our knowledge, this is the fifth report of a pathogenic missense variant through this highly conserved acidic motif of the *WNK1* in patients manifesting with GS [7,9,10,11]. Regarding the acidic motif, a recent study highlighted that all variants in exon 7, located within the acidic motif (between positions 631 and 636 of the long WNK1 isoform protein), were predicted to be pathogenic (Glu630Lys, Ala634Thr, Asp635Glu, Asp635Asn, Gln636Glu, Gln636Arg) [7]. Where four of the six missense variants modified the amino-acid charge (Glu630Lys, Asp635Asn, Gln636Glu, Gln636Arg) and seemed to be associated with a more severe phenotype [7]. However, the substitution of aspartic acid by a glutamic acid (Asp635Glu) might explain a milder phenotype observed in this particular case [7]. In addition, a similar pathogenic classification was made for other missense variants found at exon 7 of *WNK1* (Glu630Lys) [9,10]; Ala634Gly [9]; Asn635Asp [11]). For all this, the highly conserved acidic motif of *WNK1* is thought to play a fundamental structural and functional role in the function of the protein. Finally, the new variant in this family (Glu630Gly) is co-segregated with the disease in all affected members, but it is absent in healthy individuals of the family, also supporting the putative pathogenicity of the variant. Although functional studies should be performed to obtain information on the impact of this missense variant on epithelial Na^+^ channel (ENaC) function, it deserves reporting some considerations regarding its possible role in the patient’s phenotype.

Phenotypic variability includes age at diagnosis, which may vary from the first few weeks of life to mid-adulthood, the variable expression of hypertension, and the degree of sensitivity to thiazide diuretics. Since GS is an extremely infrequent disease with heterogeneous clinical manifestations, patients with mild GS can be underdiagnosed. Subjects carrying missense mutations in the *WNK1* conserved acidic motif have a clear electrolyte phenotype without hypertension, especially in comparison with those who have similar nucleotide variations in a similar protein motif of *WNK4* [9]. Interestingly, it has been shown that a less severe form of GS featuring only hyperkalemia was caused by missense mutations in the *WNK1* acidic domain [7,11]. On the other hand, in patients with the *WNK1* variants (in the acidic motif), the acidosis seemed to be more pronounced and to have an earlier onset than intron 1 deletion [9]. In addition, growth failure was linked in 28% of cases with the acidic motif and in 11% with intron 1 deletion [9].

Short stature has also been reported in some patients with GS [12,13,14,15,16,17,18,19], although the pathophysiology is poorly understood. It has been suggested that growth failure may have been due to chronic hyperkalemia and/or acidemia rather than the genetic abnormalities themselves [9,16,17]. In our family, both children (cases IV-3 and IV-8) did not have short stature, though they carried the identical *WNK1* variant (p.Glu630Gly) as did the other four members of the family. A possible explanation for the dramatic difference in phenotypes between the adult and childhood-affected family members could be the effectiveness of an early diagnosis and treatment in both children. Thus, these findings may support the hypothesis above, regarding growth failure.

There are only a few reports describing the clinical course and management of GS during pregnancy [17,20,21,22,23,24,25], where reports of preeclampsia in pregnant women with GS are so exceptional. As occurred in our patients (cases III-1 and III-8), pregnancy complicated with superimposed preeclampsia may complicate the hyperkalemia and acidosis of GS possibly due, at least in part, to increased serum progesterone concentrations with the progression of pregnancy, resulting in an anti-aldosterone effect. In GS, Na^+^/Cl^−^ co-transporter (NCC) and ENaC activity is enhanced, paracellular Cl^−^ reabsorption is further increased, and renal outer medullary K^+^ channel (ROMK) activity is further decreased, leading to increased salt reabsorption and decreased K^+^ excretion, and therefore to hypertension and hyperkalemia, respectively. Furthermore, the reduced K^+^ excretion may be also due to the increased Na^+^ reabsorption through NCC, which leaves less Na^+^ to be exchanged for K^+^ in more distal parts of the renal tubule via the conjugated functions of ENaC and ROMK, making the lumen less negative and driving less K^+^ in it through the already fewer ROMKs. The lumen is also less negative due to the increased Cl^−^ reabsorption. H^+^ secretion would be similarly impaired, thus explaining the metabolic acidosis. In addition, because of increased volume expansion, suppressed renin secretion with normal PA levels is found.

On the other hand, pregnancy causes a number of physiological changes within the renal and endocrine systems. The GFR increases and the renin–angiotensin–aldosterone system (RAAS) is upregulated. The activation of the RAAS is the cornerstone mechanism for gestational plasma volume expansion, a critical adaptation to ensure adequate placental blood flow to the fetus [26]. In GS complicated with preeclampsia (e.g., in the absence of glomerular hyperfiltration), the impaired activity of this last mechanism could aggravate K^+^ efflux, worsening hyperkalemia and acidosis. In addition, maternal metabolic acidosis could impair fetal bone growth and development, and even lead to fetal loss, as occurred in the case III-8. Our case III-1 was a unique patient with superimposed preeclampsia that was complicated by HUS with acute renal failure needing hemodialysis. Therefore, if a pregnant woman develops hypertension (superimposed preeclampsia) with unexplained hyperkalemia and acidosis, particularly in a suspicious family, GS should be suspected, and both the mother and her infant should be monitored carefully [25]. Low FE_K_^+^, FE_Na_^+^, and FE_Cl_^−^ values may establish the diagnosis, and genotype confirmation should be performed.

As we observed, in this family, treatment with low doses of thiazide and/or indapamide seems to correct all manifestations of GS because thiazide and indapamide may inhibit Na^+^, Cl^−^ cotransporter (NCC) [27], and reduce salt reabsorption, leaving more Na^+^ to be exchanged for K^+^, which is probably excreted through the flow-stimulated maxi-larger K^+^ channel (BK). In fact, thiazide and indapamide used in hypertension and presumably in GS may cause initial diuresis with hypovolemia, but, after several weeks, this diuresis ceases, and a milder state of hypovolemia persists. To explain the continuous benefit on the blood pressure, peripheral resistance may remain low. On the other hand, indapamide has a dual mechanism of action: a diuretic effect at the level of the distal tubule in the kidney and a direct vascular effect, both of which may contribute to the antihypertensive efficacy of the drug. The retard formulation contains a hydrophilic matrix, which delivers a smoother pharmacokinetic profile. This avoids unnecessary plasma peak concentrations, which may be associated with side effects [28]. Therefore, thiazide and indapamide could increase Na^+^ delivery to the ENaC to stimulate K^+^ secretion. However, another possible explanation for the control of plasma K^+^ by thiazide and indapamide in GS may be the consequence of the arginine vasopressin (AVP) released acutely by the hypovolemia, and then possibly chronically by continuing mild underhydration, which might contribute to the increase in K^+^ secretion. Finally, this is the first series of individuals with GS after long-term follow-up, and who were successfully treated with low doses of thiazide or indapamide retard.

## 5. Conclusions

In summary, we report a family of GS with a novel pathogenic missense variant in the conserved acidic motif of *WNK1*. In addition to the commonly reported characteristics of GS such as familial hyperkalemia, mild metabolic acidosis, and low PR activity with normal PA levels, our patients with the p.Glu630Gly variant of *WNK1* showed hypertension (with superimposed pre-eclampsia in two cases), short stature and low weight in adults, and initial isolate hyperkalemia in one of the children. One wide-range gene panel of next-generation sequencing in nephropathies will be very useful for differential diagnosis and categorizing subgroups of patients with inherited nephropathies, including individuals with GS; this fact may help to allow the discovery of new molecular pathways that regulate blood pressure and electrolyte homeostasis, and the development of new drug targets for treatment, as has been previously suggested [29].

## Figures and Tables

**Figure 1 genes-14-01878-f001:**
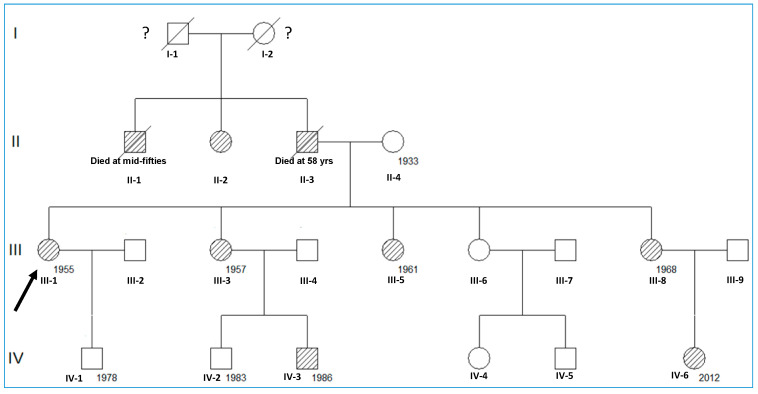
Schematic representation of the family’s pedigree. Pedigree of the family composed of 7 affected living members with GS, manifesting renal hypertension and electrolyte balance/metabolism alteration (lined). The arrow indicates the proband, and the numbers mean the year of birth.

**Figure 2 genes-14-01878-f002:**
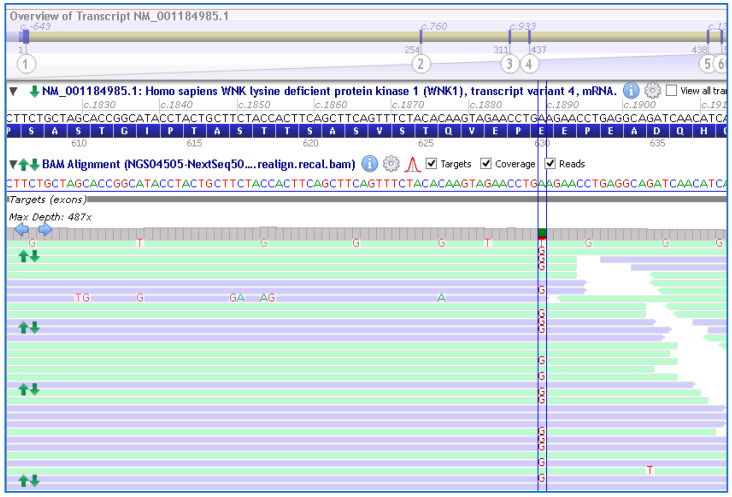
Schematic representation of the variant at exon 7; c.1889A>G (between lines) in the *WNK1* gene found in the proband and other members of the family by the NGS approach. NM_001184985.1 and GRCh38 were used.

**Figure 3 genes-14-01878-f003:**
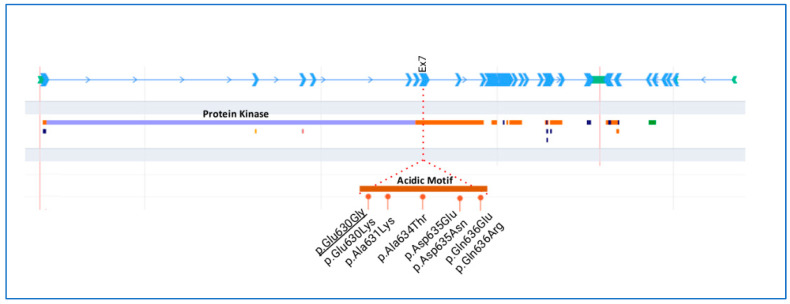
Schematic representation of the *WNK1* gene and protein. Detailed region of the acidic motif with the missense variants described in Gordon syndrome. Underlined variant was detected in the family reported herein.

**Table 1 genes-14-01878-t001:** Clinical and biochemical data of 6 cases in the family with Gordon syndrome.

Variable	III-1	III-3	III-5	III-8	IV-3	IV-6
Sex	F	F	F	F	M	F
Age at diagnosis (years)	23	30	31	43	10	<1
Age at workup (years)	61	59	50	45	27	5
Height (cm)	150	153	152	156	180	104
Weight (kg)	53	60	61	55	77	19
BMI (kg/m^2^)	23.6	25.6	26.4	22.6	24	17.6
Hypertension	Yes	Yes	Yes	Yes	Not	Not
	(preeclampsia)	2 normal pregnancies		2 abortions (preeclampsia)		
Hyperkalemia	Yes	Yes	Yes	Yes	Yes	Yes
Hyperchloremia	Yes	Yes	Yes	Yes	Yes	Yes
Metabolic acidosis	Yes	Yes	Yes	Yes	Yes	Yes
FE_K_^+^ % (normal: ≤6)	4.4	6	6	5.6	5.5	5.8
FE_Na_^+^ (normal: ≤1%)	1	0.9	0.8	0.8	1.0	0.7
FE_Cl_^−^ (normal: ≤1.2%)	1.2	1.2	0.8	0.7	1.2	1.0
eGFR, mL/min/1.73 m^2^	79	90	93	80	115	120
Plasma renin activity (normal range: 1.5–5.7 ng/mL/h)	1.5	0.1	1.6	0.1	0.2	0.65
Plasma aldosterone (normal range: 40–310 pg/mL)	193	98	145	47	208	5.8
Treatment	Indapamide	Indapamide	Indapamide	Indapamide	Indapamide	Thiazide

Abbreviations: FE_K_^+^: fractional potassium excretion; FE_Na_^+^: fractional sodium excretion; FE_Cl_^−^: fractional chloride excretion.

**Table 2 genes-14-01878-t002:** Various in silico predictors of missense deleteriousness included in the analytical pipeline.

Predictor	CADD (Damaging > 14)	DANN (0–1)	SIFT (Damaging < 0.06)	Polyphen2 (Benign < 0.03)	Vest (Possible Damaging 0.17–0.65)	EIGEN (0–1)	Mut. Assessor (Damaging > 1.8)	FATHMM-MKL (0–1)
NM_001184985.1:c.1889A>G (p.Glu630Gly)	24.3	0.9993	0, 0, 0.095	0.994	0.627	0.785	2.68	0.9807
Criteria	Ps	Ps	Ps	Ps	Ps	Ps	Ps	Ps

Ps; Pathogenic supporting.

## Data Availability

All data generated or analyzed during this study are included in this article.

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
