# Peer review of "A Spanish Family with Gordon Syndrome Due to a Variant in the Acidic Motif of WNK1"

_genes, 2023, doi:10.3390/genes14101878_

Round 1

Reviewer 1 Report

Peces and colleagues identify an ultra-rare missense variant in WNK1 associated with Gordon syndrome and segregating in an affected multigenerational family. The variant resides in a short conserved acidic domain which is a hotspot for Gordon syndrome/familial hyperkalemic hypertension. Two additional patients were previously reported to have unique variants affecting the same amino acid residue. Together, the data indicate that the variant is the most likely cause of GS in this family.

Major comments

1. Abstract – For the conclusive statement, it would be good to point out that identification of the ultra-rare novel missense variant segregating in the family helps to solidify the importance missense variants in the WNK1 acidic domain on electrolyte balance/metabolism.

1. Figure 1 – Add genotypes for the family members that were genotyped. For first generation family members, is anything known about their GS status or genotype? Presumably, the allele was passed down from one of the parents. If nothing is known, consider removing that generation as it is uninformative. For 2nd generation members, is the year of birth (other than II-4) or age at death known?

2. 2.2 Genetic studies – Add the human reference genome used (i.e. GRCh38). State the allele frequency threshold that was used for variant filtering. Various in silico predictors of missense deleteriousness were included in the analytical pipeline. Was a threshold for one or more (i.e. CADD or other) predictors used during variant filtering? State which tool and the threshold applied for filtering.

3. 3.2 Genetic findings, line 161-170 – Move the statement of the variant allele frequency (i.e. absent from gnomAD) from the discussion to this results section. Provide the CADD score for the family missense variant. Were any variants passing variant filtering identified in other known GS genes? Consider deleting the conclusive phrase (line 169-170, “which seemed to be responsible for GS inheritance in this family”) from this results section.

4. Amino acid abbreviations – Use 1- or 3- letter aa abbreviations consistently throughout the manuscript. HGVS guidelines suggest using 3-letter.

5. The ACMG clinical classification of the family variant, NM_001184985.1:c.1889A>G; p.(Glu360Gly), should be “likely pathogenic” without functional data or additional probands with the same nt variation. The authors should change “pathogenic” to “likely pathogenic” or provide an explanation.

6. Figure 2 – Suggest replacing current Figure 2 with a 2-dimensional protein schematic of WKN1 with conserved domains indicated and all heterozygous missense variants reported for GS mapped to it. This will help to drive home the importance of the acidic domain. The number of probands for each variant should also be indicated.

7- Discussion – Suggest moving the information in sentences 2-3 (lines 178-184) to the results section. E630K is mentioned three times and it should be clarified whether there are one or more probands with the variant. In general, the first paragraph is very lengthy and could be cleaned up and better organized. There is a statement that the acidic domain is important for protein function but no reference(s) are provided. It would be good to summarize existing experimental evidence from in vitro studies and mouse models. Consider mentioning the gain-of-function disease mechanism Perhaps phenotypic variability could be split out of the first paragraph. The authors should discuss possible explanations for the dramatic difference in phenotypes between the adult and childhood affected family members. The statement on lines 201-204 is author opinion but a reference has been erroneously provided. The range of diagnostic age is stated as “to late adulthood” but all affected family members were diagnosed before the age of 45 years; “adulthood” or mid-adulthood” might be more appropriate than “late adulthood.” The sentence on lines 214-216, and the last two Discussion paragraphs, could be better worded. In the last paragraph, spell out NCC and BK.

8. Conclusions – Since you are reporting a GS family with a new variant for an established GS gene, and most cases of GS appear to be explained by variants in four known genes, the second half of the last sentence (lines 238-240) does not seem to apply as written.

Minor comments

1. The therapeutic, Indapamide, should be used with a lower-case I (indapamide) throughout the manuscript.

2. Abstract – The sentence on lines 25-26 should be better worded, especially use of the term “carry out.” The second to last sentence indicates that the affected family members had late age of onset but two were children.

3. Figure 1 legend – There is a typo, “ear” should be “year.”

4. Line 139 – Two words are missing “The patient was lost to follow-up….”

5. Line 150-151 – clarify that “p10-p25” indicates percentiles.

6. Line 158 – Suggest editing “Currently, all patients were followed up for >10 years and did not feel unwell during this period” to “Currently, all patients have been followed for >10 years and remain in good health on treatment.”

7. Line 186 – Check the phrasing “acidic motif of the WKN1 gene leading in GS patients…”

8. Lines 187-190 – Add a reference.

The manuscript could use heavy editing for organization, sentence structure, word choices, and some English language.

Author Response

Thanks to the referee for such grateful comments. We are appreciated

Major comments

  1. Abstract – For the conclusive statement, it would be good to point out that identification of the ultra-rare novel missense variant segregating in the family helps to solidify the importance missense variants in the WNK1 acidic domain on electrolyte balance/metabolism.

Ok Accepted, a new sentence in the abstract has been added regarding the point above cited.

Figure 1 – Add genotypes for the family members that were genotyped. For first generation family members, is anything known about their GS status or genotype? Presumably, the allele was passed down from one of the parents. If nothing is known, consider removing that generation as it is uninformative. For 2nd generation members, is the year of birth (other than II-4) or age at death known?

Ok Accepted, information regarding II generation availability and GS status have been included in the legend of figure 1

  1. 2.2 Genetic studies – Add the human reference genome used (i.e. GRCh38). State the allele frequency threshold that was used for variant filtering. Various in silico predictors of missense deleteriousness were included in the analytical pipeline. Was a threshold for one or more (i.e. CADD or other) predictors used during variant filtering? State which tool and the threshold applied for filtering.

Ok accepted. GRCh37 (hg19) was used as the human genome of reference. The allele frequency threshold used for variant filtering was 0.01 to avoid polymorphism higher than 1% and has been also stated within the text. In addition, the threshold for predictors used during variant filtering was also stated.

  1. 3.2 Genetic findings, line 161-170 – Move the statement of the variant allele frequency (i.e. absent from gnomAD) from the discussion to this results section.

OK accepted the sentence has been moved to the result section

Provide the CADD score for the family missense variant.

OK, This CADD-PFred score was 24.3. Scores above 20 are predicted to be among the 1.0% most deleterious possible substitutions in the human genome.

Were any variants passing variant filtering identified in other known GS genes?

None

Consider deleting the conclusive phrase (line 169-170, “which seemed to be responsible for GS inheritance in this family”) from this results section.

Ok the phrase has been rephrased

  1. Amino acid abbreviations – Use 1- or 3- letter aa abbreviations consistently throughout the manuscript. HGVS guidelines suggest using 3-letter.

Ok, accepted HGVS guidelines were used

  1. The ACMG clinical classification of the family variant, NM_001184985.1:c.1889A>G; p.(Glu360Gly), should be “likely pathogenic” without functional data or additional probands with the same nt variation. The authors should change “pathogenic” to “likely pathogenic” or provide an explanation.

Ok, accepted, this family variant has been re-stated to be LP.

  1. Figure 2 – Suggest replacing current Figure 2 with a 2-dimensional protein schematic of WKN1 with conserved domains indicated and all heterozygous missense variants reported for GS mapped to it. This will help to drive home the importance of the acidic domain. The number of probands for each variant should also be indicated.

Ok, we reconsider to generate a new figure; Figure 3 with these suggestions.

7- Discussion – Suggest moving the information in sentences 2-3 (lines 178-184) to the results section.

Ok done,

E630K is mentioned three times and it should be clarified whether there are one or more probands with the variant.

Ok, we tried

In general, the first paragraph is very lengthy and could be cleaned up and better organized.

Ok the paragraph has been re-stated

There is a statement that the acidic domain is important for protein function but no reference(s) are provided.

Ok

It would be good to summarize existing experimental evidence from in vitro studies and mouse models.

Ok

Consider mentioning the gain-of-function disease mechanism

Perhaps phenotypic variability could be split out of the first paragraph.

Ok

The authors should discuss possible explanations for the dramatic difference in phenotypes between the adult and childhood affected family members.

Ok done,

The statement on lines 201-204 is author opinion but a reference has been erroneously provided.

Ok that´s correct

The range of diagnostic age is stated as “to late adulthood” but all affected family members were diagnosed before the age of 45 years; “adulthood” or mid-adulthood” might be more appropriate than “late adulthood.”

Ok accepted.

The sentence on lines 214-216, and the last two Discussion paragraphs, could be better worded.

Ok

In the last paragraph, spell out NCC and BK.

Ok done

  1. Conclusions – Since you are reporting a GS family with a new variant for an established GS gene, and most cases of GS appear to be explained by variants in four known genes, the second half of the last sentence (lines 238-240) does not seem to apply as written.

Minor comments

The therapeutic, Indapamide, should be used with a lower-case I (indapamide) throughout the manuscript.

OK, accepted

  1. Abstract – The sentence on lines 25-26 should be better worded, especially use of the term “carry out.” The second to last sentence indicates that the affected family members had late age of onset but two were children.

Ok done, it has been rephrased

  1. Figure 1 legend – There is a typo, “ear” should be “year.”

Ok done,

  1. Line 139 – Two words are missing “The patient was lost to follow-up….”

Ok done,

  1. Line 150-151 – clarify that “p10-p25” indicates percentiles.

Ok done,

  1. Line 158 – Suggest editing “Currently, all patients were followed up for >10 years and did not feel unwell during this period” to “Currently, all patients have been followed for >10 years and remain in good health on treatment.”

Ok, accepted.

  1. Line 186 – Check the phrasing “acidic motif of the WKN1 gene leading in GS patients…”

Ok done.

  1. Lines 187-190 – Add a reference.

Ok done

Comments on the Quality of English Language

The manuscript could use heavy editing for organization, sentence structure, word choices, and some English language.

 The manuscript has been English revised

Reviewer 2 Report

This is a good article in every way. A detailed description of one of the variants of familial hypertension of Gordon's syndrome is given. The authors found a novel heterozygous missense variant in the WNK1 gene and gave a detailed description of the phenotypic consequences. The most optimal method of treatment was also proposed. The article is of undoubted interest for specialists in the field of medical genetics and for clinical medicine.

Author Response

This is a good article in every way. A detailed description of one of the variants of familial hypertension of Gordon's syndrome is given. The authors found a novel heterozygous missense variant in the WNK1 gene and gave a detailed description of the phenotypic consequences. The most optimal method of treatment was also proposed. The article is of undoubted interest for specialists in the field of medical genetics and for clinical medicine.

Thanks to the referee for such grateful comments. We are appreciated

Reviewer 3 Report

Dear Authors

Figure 1: There is only the numbering of generations, the numbers of patients and relatives are missing.

Lines 199-200  : “(E630G) co-segregated with the disease in all affected members, but it is absent in healthy individuals of the family”. It is not clear how many unaffected relatives were analyzed. It would be better if all analyzed persons were marked in this figure (eg Glu/Gly and Glu/Glu).

In Materials and Methods, lines 95-98 contain a list of in silico tools, but any results from their use is absent in the Result section; line 180-181:  “It is also predicted to be pathogenic by several in silico bioinformatic tools”. Does this mean that all methods showed pathogenicity or not?

Line 132, I fink “first sister” would be more correct than “older sister” because the proband is the older sister

Genetic findings (beginning in line 161) should contain the list of pathogenicity criteria according to [8]

Line 189: term “L-WNK1” appeared, but was not explained

Line 210: "similar WNK4 mutations" is incorrect; possibly "WNK4 mutations in the similar motif".

Lines 217-218: one of the sentences is redundant

Mistakes:

Line 71  “ear” should be replaced with “year”

Line 135 “individuals” should be replaced with “individual”

Author Response

Thanks to the referee for such grateful comments. We are appreciated

Dear Authors

Figure 1: There is only the numbering of generations, the numbers of patients and relatives are missing.

Ok the figure 1 has been changed to include this suggestion.

Lines 199-200 : “(E630G) co-segregated with the disease in all affected members, but it is absent in healthy individuals of the family”. It is not clear how many unaffected relatives were analyzed. It would be better if all analyzed persons were marked in this figure (eg Glu/Gly and Glu/Glu).

This will be explained in the text.

In Materials and Methods, lines 95-98 contain a list of in silico tools, but any results from their use is absent in the Result section; line 180-181:  “It is also predicted to be pathogenic by several in silico bioinformatic tools”. Does this mean that all methods showed pathogenicity or not?

Ok, this aspect has been modified.

Line 132, I think “first sister” would be more correct than “older sister” because the proband is the older sister

Ok, it has been corrected

Genetic findings (beginning in line 161) should contain the list of pathogenicity criteria according to [8].

Ok accepted it has been included.

Line 189: term “L-WNK1” appeared, but was not explained

Ok it has been explained. L-WNK1 means Long-WNK1

Line 210: "similar WNK4 mutations" is incorrect; possibly "WNK4 mutations in the similar motif".

Ok that´s correct, it has been changed.

Lines 217-218: one of the sentences is redundant

Ok it has been changed. 

Mistakes:

Line 71  “ear” should be replaced with “year”

Ok it has been changed.

Line 135 “individuals” should be replaced with “individual”

Ok it has been changed.